# Association of Carotenoids Concentration in Blood with Physical Performance in Korean Adolescents: The 2018 National Fitness Award Project

**DOI:** 10.3390/nu12061821

**Published:** 2020-06-18

**Authors:** Dawn Jeong, Saejong Park, Hyesook Kim, Oran Kwon

**Affiliations:** 1Department of Clinical Nutrition Science, the Graduate School of Clinical Health Sciences, Ewha Womans University, 52, Ewhayeodae-gil, Seodaemun-gu, Seoul 03760, Korea; doll1020@naver.com; 2Department of Sport Science, Korea Institute of Sport Science, Seoul 03760, Korea; saejpark@kspo.or.kr; 3Department of Nutritional Science and Food Management, Ewha Womans University, 52, Ewhayeodae-gil, Seodaemun-gu, Seoul 03760, Korea

**Keywords:** blood carotenoids, physical performance, Korean adolescents

## Abstract

Adolescent physical performance is not only dependent on exercise but also on the role of antioxidants obtained through a healthy diet. However, few studies have specifically identified the relationship between carotenoids, a common antioxidant, and physical performance. This cross-sectional study aims to investigate the association between the level of carotenoids in the blood and physical performance among Korean adolescents aged 13 to 18 years. The study sample consisted of 450 participants (190 males, 260 females) from the 2018 National Fitness Award project. In boys, multiple regression analysis after adjustment for age, body mass index (BMI), smoking, drinking, and physical activity revealed that the α-carotene level was positively associated with a 20-m progressive aerobic cardiovascular endurance run (PACER) (*β* = 5.350, *p* < 0.05) and estimated maximal oxygen consumption (VO_2max_) (*β* = 1.049, *p* < 0.05). In girls, after adjustment for age, BMI, smoking, drinking, and physical activity, the levels of α-carotene were positively associated with a 20-m PACER (*β* = 3.290, *p* < 0.05), VO_2max_ (*β* = 0.644, *p* < 0.05) and curl-up (*β* = 5.782, *p* < 0.05), and β-carotene (*β* = 2.983, *p* < 0.05) and total carotenoids (*β* = 4.248, *p* < 0.05) were positively associated with curl-up. Our results suggest that an increased level of carotenoids in the blood may be associated with better physical performance among adolescents in Korea.

## 1. Introduction

Adolescence is an important period for physical and mental development. It is also an active time of physical growth and development in the human body [1]. The physical conditions formed during adolescence affect various areas of a person’s life and greatly influence the quality of life of these individuals as adults [2]. Physical performance can be seen as an important factor in youth health, given its association with cardiometabolic risk [3], obesity [4], and metabolic syndrome [5]. Besides its relevance to physical health, physical performance affects academic achievement, cognitive function, and other important skills needed during adolescence [6,7].

Physical performance is a set of attributes linked to a person’s ability to perform physical activities that require aerobic capacity, endurance, strength, or flexibility, and is mainly determined by lifestyle factors, including physical activity, as well as genetically inherited ability [8]. Nutritional status, especially antioxidant nutrition status associated with eating habits, is considered to be one of the most important factors related to physical performance [9]. An individual’s physical performance may be related to an oxidative stress state. Hence, to improve physical performance, it is necessary to increase the antioxidant state. Physical activity such as exercise increases endogenous antioxidant capacity [10], and the consumption of antioxidant substances through foods, such as fruits and vegetables, can increase the exogenous antioxidant capacity [11]. A study conducted in Spain revealed that blood levels of most antioxidant vitamins, such as vitamin C, α-tocopherol, and β-carotene, were positively associated with cardiorespiratory and muscular fitness [12].

Carotenoids are pigments found in nearly all colored fruits and green leafy vegetables. Nearly 600 different types exist in nature [13], and only about 50 carotenoids are found in a typical human diet [14]. Humans do not synthesize carotenoids. Instead, carotenoids must be ingested in food or via supplementation. Only 20 carotenoids have been found in human tissues and blood samples, with β-carotene, β-cryptoxanthin, lutein, zeaxanthin, and lycopene generally being the most abundant [15]. Carotenoids have been associated with various health benefits, and, mechanistically, the primary benefits of carotenoids can be explained by their antioxidant potential [16]. Several lines of evidence highlight that carotenoids can decrease oxidative stress [16,17,18,19].

Although carotenoids appear to be associated with lowering oxidative stress and possibly improve physical performance, to the best of our knowledge, studies on the relationship between carotenoid level and physical performance are still insufficient. Such information would aid health and nutrition promotion among people. There have also been few studies examining such associations during adolescence, a critical period considering that the physical performance level in adolescence can affect post-adult diseases. Therefore, in this cross-sectional study, we evaluated the association between the carotenoid levels in the blood and physical performance in adolescents aged 13–18 years.

## 2. Materials and Methods

### 2.1. Study Population

This cross-sectional study was performed based on the National Fitness Award project 2018 for adolescents in South Korea. The National Fitness Award project is a large-scale national project currently managed by 21 centers [20]. The Korea Institute of Sports Science has been conducting this project for youth, adults, and seniors since 2012 with the support of the Ministry of Culture, Sports, and Tourism to promote health by exercise, physical, and sporting activities in daily life. Among a total of 559 adolescents aged 13–18 years, we excluded participants who (1) had no analysis on carotenoid levels by drawing blood (*n* = 57); (2) were missing data on covariates (*n* = 30); and (3) had no data on their physical performance (*n* = 22), such as handgrip strength, a 20-m progressive aerobic cardiovascular endurance run (PACER), estimated maximum oxygen consumption (VO_2max_), curl-up, and sit-and-reach. Finally, the study sample included a total of 450 participants (190 men and 260 women) aged 13–18 years. This study received approval from the Institutional Review Board (IRB) of the Korea Institute of Sport Science, and Ewha Womans University. All individuals provided informed consent before enrollment.

### 2.2. General Characteristics

Study participants were interviewed by trained interviewers to obtain general information on health-related behaviors, including eating habits, smoking status, alcohol consumption, and physical activity. Current smokers were defined as participants who reported smoking more than one cigarette in the last month. Current drinkers were defined as participants who reported drinking more than one cup of alcohol in the last month. Physical activity was defined as engaging in at least one of the four intensity levels (high, moderate, cardio-intensive, and strength-intensive) at least once a week. Questions regarding eating habits focused on the consumption of breakfast, fruits, fast foods, and carbonated drinks, with “frequent consumption” being defined as eating the food once a month in recent times. Current drinkers and smokers were defined as consuming alcohol or smoking more than once a month in recent times.

### 2.3. Measurement of Anthropometry

Anthropometry parameters were measured by specialized medical staff. The standing height of the participant was measured to the nearest 0.1 cm by using a stadiometer (Seca, Seca Corporation, Columbia, MD, USA). Body weight was measured to the nearest 0.1 kg with an electronic weight scale (Inbody 720, Biospace, Seoul, Korea). Body mass index (BMI) was calculated as weight divided by the square of the height (kg m^−2^).

### 2.4. Measurement of Blood Carotenoid Levels

After the participants fasted for 8 h, blood samples were collected and plasma was obtained immediately by centrifugation at 3000 rpm for 3 min. The α-Carotene, β-carotene, β-cryptoxanthin, lutein, zeaxanthin, and lycopene were determined using an HPLC instrument (Shiseido Co., Ltd., Tokyo, Japan) equipped with a photodiode array (Shiseido Co. Ltd., Tokyo, Japan) and a YMC C30 column (5 μM, 4.6 × 250 mm; YMC Europe GmbH, Dinslaken, Germany). Total carotenoids (μmol L^−1^) were calculated as the sum of α-carotene, β-carotene, β-cryptoxanthin, lutein, zeaxanthin, and lycopene.

### 2.5. Assessment of Physical Performance

All the parameters of physical performance were measured by certified health and fitness instructors. The assessment of physical performance for adolescents included the handgrip strength test, the 20-m PACER test, the VO_2max_ test, the curl-up test, and the sit-and-reach test. The National Fitness Award project test items for adolescents showed a high consistency, with reliability ranging from 0.87–0.99 [21]. The specific methods for physical performance assessments were as follows:Handgrip strength (kg, %): Handgrip strength is a measure of muscular strength determined using a hand dynamometer (GRIP-D 5101, Takei, Niigata, Japan). Participants must hold the handle of the handgrip with the second finger. If the handle does not fit, it must be adjusted accordingly with the screw. Participants are asked to outstretch their arms and pull as hard as they can while keeping their body and arm at a 15° angle. Then, the hand dynamometer is held with maximum force. This position is maintained for 5 s. The grip force is measured twice, alternating between left and right, with the highest value recorded to the nearest 0.1 kg. Relative handgrip strength (%) is calculated by dividing the absolute handgrip strength (kg) by the participant’s body weight (kg) and multiplying the number by 100.The 20-m PACER (number of times): PACER is a method of measuring aerobic capacity. Each lane on the 20-m course is divided, and a line is marked at the end of 20 m with tape. At the “start” signal, the participants run across a 20-m distance before they hear the beep from the recorded compact disk (CD). The participants who reach the opposite line before the signal is sounded must wait for the signal to sound. When the signal sounds, the participant runs toward the end of the opposite line. If the line has not been reached before the tone, the participant may take one reverse run when the signal is sounded. However, if the line is not reached before the second beep, the participant will be eliminated. In the same way, before the second beep, the participant must continue to run accordingly until the line is not reached. The maximum number of runs for each participant is recorded.Estimated VO_2max_ (mL∙kg^−1^∙min^−1^): VO_2max_ is the most accurate method for measuring cardiopulmonary endurance with maximum oxygen intake. However, due to VO_2max_’s equipment costs and vigorous measurements, PACER is the most widely used test for youth physical fitness tests. The estimated VO_2max_ was calculated using a quadratic model based on the 20-m PACER measurement [22].Curl-up (number of times): The curl-up is a way to measure muscle endurance. First, participants must lie down with their back on the mat, knees bent, and their feet placed about 30 cm away from their hips. Both arms should be placed crosswise on the chest. With the feet fixed, participants must raise their upper body at the “start” signal so that each elbow touches the corresponding thigh. This movement is counted as one curl-up. To perform more curl-ups, the steps are repeated. The number of curl-ups is then measured and recorded.Sit-and-reach (cm): The sit-and-reach is a measure of flexibility. Participants must remove their shoes and sit upright with their legs straightened so that the soles of both feet are in full contact with the vertical plane of the measuring instrument. The distance between the feet should be a maximum of 5 cm. The participant is prepared with both hands stretched out over the measurer. The upper body must be kept down and extended as far forward as possible without bending the knees. The sit-and-reach is performed twice, and the highest measurement is recorded to the nearest 0.1 cm.

### 2.6. Statistical Analysis

The general characteristics of the participants were presented as mean and standard deviation (SD) for continuous variables. Categorical variables were presented as numbers and percentages. The differences in the distribution of categorical variables (smoking, drinking, physical activity) between boys and girls were analyzed using the Chi-square test. Differences in means of the general characteristics, blood levels of carotenoids, and physical performance between boys and girls were analyzed using the Student’s *t*-test. Associations between plasma carotenoid levels and physical performance were analyzed through the multiple linear regression analysis after adjusting for covariates. Model 1 was adjusted for age. Model 2 was adjusted for age and BMI, and Model 3 was adjusted for age, BMI, and physical activities. All statistical analyses were performed using SPSS (version 18.0; SPSS, Inc., Chicago, IL, USA). Statistical significance was set at *p* < 0.05.

## 3. Results

### 3.1. General Characteristics of the Participants

The mean age was 15.3 ± 2.0 years for the boys and 14.8 ± 2.0 years for the girls, respectively (Table 1). The age, height, weight, and BMI were significantly higher in boys than in girls. The percentages of physically active participants and current smokers were also higher in boys than in girls.

### 3.2. Carotenoid Levels of the Participants

The mean lutein level was significantly higher (*p* < 0.01) in girls (0.26 ± 0.11 μmol L^−1^) than in boys (0.23 ± 0.10 μmol L^−1^), whereas the mean zeaxanthin level (*p* < 0.0001) was higher in boys (0.20 ± 0.07 μmol L^−1^) than in girls (0.17 ± 0.08 μmol L^−1^). The mean level of total carotenoids (the sum of α-carotene, β-carotene, β-cryptoxanthin, lutein, zeaxanthin, and lycopene) was higher in girls than in boys, but the difference was not significant (Table 2).

### 3.3. Physical Performance Levels of the Participants

The mean values of absolute and relative handgrip strength, 20-m PACER, estimated VO_2max_, and curl-up were significantly higher in boys than in girls. Meanwhile, the mean value of sit-and-reach was higher in girls than in boys (Table 3).

### 3.4. Association between Blood Carotenoids and Physical Performance

As shown in Table 4, after adjustment for age (Model 1), the multiple regression analysis revealed that the α-carotene level was positively associated with the 20-m PACER and estimated VO_2max_ in boys. This trend was also observed after further adjustment (Model 3) for BMI, smoking, drinking, and physical activity (*β* = 5.350, *p* < 0.05 for 20-m PACER; *β* = 1.049, *p* < 0.05 for VO_2max_). In the age-adjusted model (Model 1), the total carotenoid level was positively associated with relative handgrip strength (*β* = 3.824, *p* < 0.05), 20-m PACER (*β* = 5.327, *p* < 0.05), and estimated VO_2max_ (*β* = 1.853, *p* < 0.01) in boys. However, this association was not significant after further adjustment (Model 3). Similar to the results in boys, the α-carotene level in blood was positively related to 20-m PACER (*β* = 3.290, *p* < 0.05), estimated VO_2max_ (*β* = 0.644, *p* < 0.05), and curl-up (*β* = 5.782, *p* < 0.05) in girls (Model 3). In girls, after adjustment for age, BMI, smoking, drinking, and physical activity (Model 3), the β-carotene (*β* = 2.983, *p* < 0.05) and total carotenoids (*β* = 4.248, *p* < 0.05) levels were positively associated with curl-up.

## 4. Discussion

In this cross-sectional study, we found that the α-carotene level was positively associated with the 20-m PACER and estimated VO_2max_ in boys and girls, and the β-carotene and total carotenoids level was positively associated with curl-up in girls. To the best of our knowledge, this is the first study to show a positive association between the carotenoid levels in the blood and physical performance in adolescents.

Although unrelated to adolescents, the importance of maintaining physical performance in older adults is being emphasized as the elderly population increases worldwide, and the role of carotenoids [23] in preventing sarcopenia associated with physical performance in old age has been revealed. A study of American elderly individuals reported a relationship between low serum carotenoids and a decline in walking speed in older women [24], and a study in Italy revealed that skeletal muscle strength decreases over time (6 years) in older adults with low plasma carotenoid levels [25]. As such, previous studies have reported that carotenoids are directly or indirectly related to physical performance in the elderly.

Carotenoids are abundantly present in deeply pigmented fruits and vegetables. Orange-yellow vegetables and fruits are rich in β-carotene and α-carotene, while β-cryptoxanthin, lycopene, and lutein are found in citrus fruits, tomatoes and tomato products, and dark green vegetables, respectively [26]. Egg yolk is a highly bioavailable source of zeaxanthin and lutein [27]. Carotenoids perform several functions in human health, with positive roles in eye health, cognitive function, heart health, cancer prevention, maternal and infant nutrition, fertility, and immune modulation [18]. Although specific carotenoids may also act through additional mechanisms, the primary benefit of carotenoids can be explained by their antioxidant capacity [16]. As antioxidants, carotenoids can be very efficient physical and chemical quenchers of singlet oxygen as well as potent scavengers of other reactive oxygen species (ROS), which contribute to oxidative stress.

Oxidative stress occurs when there is an imbalance between the free radical activity and antioxidant activity, which, in turn, leads to poor muscle function and physical performance. Carotenoids, along with vitamins C and E and polyphenols (e.g., flavonoids), are well known as main exogenous antioxidants that can reduce this oxidative stress. Carotenoids act as antioxidants in the human body, helping to maintain an antioxidant–ROS balance which can help improve physical performance. Nieman et al. [28] reported that increased levels of carotenoids in the blood reduces muscle damage. In a study of Spanish adolescents, higher concentrations of vitamin C in male adolescents and β-carotene in female adolescents were positively associated with maximal oxygen consumption, and higher concentrations of β-carotene and α-tocopherol in male adolescents and β-carotene in female adolescents were associated with better performance in the standing long jump test [12]. Some studies examining the relationship between nutritional intake status and physical performance in adolescents have considered antioxidant activity as a mechanism for these associations [29,30].

A recent review focused on the potential effect of carotenoids on obesity via their direct and/or adipose tissue-driven indirect biological effects on the brain [31]. Maintaining a normal BMI with age-appropriate muscle mass is important for physical performance in adolescents [32,33]. In some observational studies, obesity has been associated with low circulating carotenoid concentrations [34,35]. The Coronary Artery Risk Development in Young Adult (CARDIA) study revealed that BMI was strongly inversely related to all measured plasma carotenoids, including α-carotene, β-carotene, β-cryptoxanthin, and lutein/zeaxanthin, except lycopene [36]. In addition, recent systematic review and meta-analysis results also emphasized that serum carotenoid concentrations are strongly inversely related to metabolic syndrome, one of the disorders associated with obesity [37]. Research in Brazil found that overweight children and adolescents presented a greater chance of a decrease in serum concentrations of carotenoids when compared with non-overweight subjects [38]. In this study, we also found a negative correlation (*p* < 0.01) between carotenoid levels in the blood and BMI in adolescents (data not shown).

Nevertheless, when compared with the blood levels of adolescents in other countries, the levels in our study participants were similar to or slightly lower than those of Norwegian adolescents [39], and slightly higher than those of US adolescents [40]. It is likely that these discrepancies arise from the different dietary habits and food intakes across different countries. As carotenoid levels in the blood of adolescents are important indicators of physical performance, further research is needed.

In this study, we do not know the exact reason why carotenoid levels in the blood were associated with different performance indicators in boys and girls, that is, cardiorespiratory in men and muscular endurance in women. However, it is purported that the in vivo action of carotenoids is caused by different aspects of sex. Carotenoids are activated by various variables within the body of men and women. The results of one study [41] showed that the total carotenoid concentration in women’s blood was higher than that in men, even though the dietary carotenoid intake was lower in women, indicating that the absorption and role of carotenoids in the body varies depending on gender. Further research is needed to examine the association between the level of carotenoids in the blood and the physical strength indicators, which have been considered as variables in understanding gender-specific blood indicators and oxidative stress [42].

Several potential limitations of our study deserve consideration. First, we cannot determine the causal relationship between carotenoids and physical performance because our research is a cross-sectional study design. Second, we did not have the data on the total antioxidant capacity in the blood, so we could not evaluate its association with physical performance due to the limited number of samples. Third, although the amount of consumed carotenoids could have affected the result, this study did not include a nutritional assessment. Fourth, our subjects were at different developmental stages, aged 13–18 years. In particular, considering the fact that hormonal metabolism may differ depending on the stage of development in girls, this may have affected the research results. In order to confirm the effects of carotenoids on physical performance, more systematically designed longitudinal studies will be needed along with the expansion of measurement indicators, such as assessments of nutrition and total antioxidant capacity. Despite these limitations, this is the first study to show a positive association between the carotenoid levels in the blood and physical performance in Korean adolescents. This finding is particularly meaningful because our study involved students in adolescence, which is a critical period as the physical performance level in adolescence can be linked to post-adult health conditions or diseases.

## 5. Conclusions

In conclusion, we found a positive association between the α-carotene level in blood and the 20-m PACER and estimated VO_2max_ in boys and girls, and between the β-carotene and total carotenoid levels and curl-up performance in girls. Our results suggest that an increased carotenoids level in the blood might be associated with improved physical performance among adolescents in Korea. It is necessary to encourage the intake of carotenoids through healthy eating habits, such as the regular consumption of fruits and vegetables, to improve the physical performance of adolescents. Further longitudinal studies are also needed to confirm the effects of carotenoids on physical performance.

## Figures and Tables

**Table 1 nutrients-12-01821-t001:** General characteristics of Korean adolescents.

Characteristic	Total (*n* = 450)	Boys (*n* = 190)	Girls (*n* = 260)	*p*
Age (years)	15.5 ± 2.0	15.31 ± 2.0	14.8 ± 2.0	<0.05
Height (cm)	164.6 ± 7.7	169.8 ± 7.3	160.8 ± 5.3	<0.0001
Weight (kg)	57.4 ± 12.0	63.1 ± 12.8	53.2 ± 9.5	<0.0001
BMI (kg m^−2^)	21.0 ± 3.5	21.8 ± 3.8	20.5 ± 3.1	<0.0001
Current smoker (*n*, %)	13 (2.9)	10 (5.3)	3 (1.2)	<0.05
Current drinker (*n*, %)	40 (8.9)	20 (10.5)	20 (7.7)	NS
Physical activity (*n*, %)	381 (84.7)	175 (92.1)	206 (79.2)	<0.0001

Data are presented as mean ± SD or number (percentage). Current smokers were defined as participants who reported smoking more than one cigarette in the last month. Current drinkers were defined as participants who reported drinking more than one cup of alcohol in the last month. Physical activity was defined as engaging in at least one of the four intensity levels (high, moderate, cardio-intensive, and strength-intensive) at least once a week. BMI, body mass index.

**Table 2 nutrients-12-01821-t002:** Blood carotenoid level of the adolescents.

Carotenoid Concentration (μmol L^−1^)	Total (*n* = 450)	Boys (*n* = 190)	Girls (*n* = 260)	*p*
α-Carotene	0.15 ± 0.06	0.16 ± 0.07	0.15 ± 0.05	NS
β-Carotene	0.52 ± 0.38	0.52 ± 0.45	0.52 ± 0.32	NS
β-Cryptoxanthin	0.52 ± 0.68	0.50 ± 0.48	0.53 ± 0.81	NS
Lutein	0.24 ± 0.11	0.23 ± 0.10	0.26 ± 0.11	<0.01
Zeaxanthin	0.19 ± 0.08	0.20 ± 0.07	0.17 ± 0.08	<0.0001
Lycopene	0.53 ± 0.30	0.53 ± 0.31	0.52 ± 0.31	NS
Total carotenoids	1.62 ± 0.88	1.60 ± 0.79	1.64 ± 0.94	NS

Data are presented as mean ± SD. Total carotenoids: the sum of the blood levels of the six individual carotenoids.

**Table 3 nutrients-12-01821-t003:** Physical performance levels of Korean adolescents.

Physical Performance Measure	Total (*n* = 450)	Boys (*n* = 190)	Girls (*n* = 260)	*p*
Absolute handgrip strength (kg)	28.5 ± 8.7	35.6 ± 7.9	23.2 ± 4.7	<0.0001
Relative handgrip strength (%)	49.8 ± 12.1	57.3 ± 11.6	44.3 ± 9.1	<0.0001
20-m PACER (reps)	35.2 ± 16.9	46.8 ± 17.0	26.8 ± 10.6	<0.0001
Estimated VO_2max_ (mL∙kg^−1^∙min^−1^)	39.3 ± 5.5	44.1 ± 4.4	35.9 ± 3.0	<0.0001
Curl-up (reps)	25.5 ± 17.6	32.9 ± 17.8	20.1 ± 15.3	<0.0001
Sit-and-reach (cm)	11.1 ± 10.2	7.9 ± 10.0	13.5 ± 9.7	<0.0001

Data are presented as mean ± SD. PACER, progressive aerobic cardiovascular endurance run; VO_2max_, maximal oxygen uptake.

**Table 4 nutrients-12-01821-t004:** Multiple linear regression analysis for the association between blood carotenoid levels and physical performance in adolescents.

		Absolute Handgrip Strength			Relative Handgrip Strength			20-m PACER			Estimated VO_2max_			Curl-up			Sit-and-Reach		
		β	SE	*p*	β	SE	*p*	β	SE	*p*	B	SE	*p*	β	SE	*p*	β	SE	*p*
**Boys**																			
α-Carotene	Model 1	−1.027	0.903	NS	−0.060	1.554	NS	6.008	2.339	<0.05	1.472	0.618	<0.05	−2.698	2.482	NS	−1.225	1.399	NS
	Model 2	−0.844	0.891	NS	−1.116	1.195	NS	5.097	2.186	<0.05	1.000	0.428	<0.05	−3.663	2.320	NS	−1.250	1.407	NS
	Model 3	−0.977	0.898	NS	−1.257	1.207	NS	5.350	2.198	<0.05	1.049	0.431	<0.05	−3.855	2.327	NS	−1.300	1.426	NS
β-Carotene	Model 1	−0.284	0.527	NS	2.262	0.890	<0.05	1.758	1.380	NS	0.880	0.360	<0.05	4.076	1.419	**<0.01**	0.385	0.816	NS
	Model 2	0.061	0.534	NS	0.368	0.716	NS	0.044	1.327	NS	0.008	0.260	NS	2.500	1.385	NS	0.374	0.843	NS
	Model 3	0.005	0.541	NS	0.303	0.727	NS	0.314	1.341	NS	0.061	0.263	NS	2.713	1.394	NS	0.415	0.858	NS
β-Cryptoxanthin	Model 1	1.144	0.752	NS	2.269	1.287	NS	2.768	1.977	NS	0.905	0.520	NS	2.407	2.072	NS	1.469	1.166	NS
	Model 2	1.393	0.742	NS	0.988	1.001	NS	1.630	1.855	NS	0.320	0.364	NS	1.260	1.955	NS	1.466	1.177	NS
	Model 3	1.310	0.755	NS	0.894	1.020	NS	1.801	1.881	NS	0.354	0.369	NS	1.071	1.979	NS	1.441	1.202	NS
Lutein	Model 1	−0.269	1.218	NS	1.951	2.084	NS	2.988	3.191	NS	1.164	0.839	NS	3.207	3.338	NS	−0.202	1.884	NS
	Model 2	0.101	1.205	NS	−0.115	1.615	NS	1.152	2.990	NS	0.227	0.586	NS	1.370	3.148	NS	−0.241	1.901	NS
	Model 3	−0.084	1.218	NS	−0.318	1.636	NS	1.592	3.017	NS	0.313	0.591	NS	1.357	3.169	NS	−0.300	1.933	NS
Zeaxanthin	Model 1	−0.034	1.162	NS	1.463	1.989	NS	3.645	3.039	NS	1.139	0.800	NS	1.310	3.190	NS	0.437	1.797	NS
	Model 2	0.238	1.146	NS	−0.048	1.536	NS	2.308	2.840	NS	0.454	0.557	NS	−0.048	2.996	NS	0.414	1.808	NS
	Model 3	0.076	1.167	NS	−0.227	1.567	NS	2.744	2.885	NS	0.540	0.565	NS	−0.069	3.037	NS	0.323	1.851	NS
Lycopene	Model 1	0.528	1.360	NS	5.373	2.264	**<0.05**	8.564	3.661	**<0.05**	2.883	0.942	**<0.01**	10.219	3.794	**<0.01**	−1.518	2.101	NS
	Model 2	1.382	1.364	NS	1.132	1.779	NS	4.956	3.539	NS	0.971	0.694	NS	6.728	3.698	NS	−1.551	2.158	NS
	Model 3	1.074	1.390	NS	0.744	1.813	NS	5.988	3.592	NS	1.173	0.704	NS	6.975	3.756	NS	−1.486	2.210	NS
Total carotenoids	Model 1	0.850	1.030	NS	3.824	1.747	**<0.05**	5.327	2.681	**<0.05**	1.853	0.701	**<0.01**	4.316	2.816	NS	1.788	1.591	NS
	Model 2	1.415	1.027	NS	0.971	1.381	NS	2.838	2.553	NS	0.557	0.500	NS	1.783	2.694	NS	1.801	1.623	NS
	Model 3	1.243	1.044	NS	0.778	1.406	NS	3.248	2.585	NS	0.637	0.507	NS	1.596	2.723	NS	1.786	1.656	NS
**Girls**																			
α-Carotene	Model 1	1.079	0.732	NS	1.649	1.452	NS	3.107	1.676	NS	0.697	0.472	NS	5.849	2.395	**<0.05**	−1.392	1.534	NS
	Model 2	1.175	0.678	NS	1.442	1.327	NS	2.936	1.605	NS	0.575	0.315	NS	5.813	2.398	**<0.05**	−1.355	1.534	NS
	Model 3	1.167	0.686	NS	1.402	1.338	NS	3.290	1.607	**<0.05**	0.644	0.315	**<0.05**	5.782	2.401	**<0.05**	−1.035	1.534	NS
β-Carotene	Model 1	0.267	0.387	NS	0.464	0.767	NS	0.052	0.889	NS	−0.012	0.250	NS	3.404	1.259	**<0.01**	0.829	0.808	NS
	Model 2	0.243	0.359	NS	0.516	0.700	NS	0.095	0.851	NS	0.019	0.167	NS	3.414	1.260	**<0.01**	0.820	0.808	NS
	Model 3	0.171	0.366	NS	0.321	0.712	NS	0.038	0.860	NS	0.008	0.169	NS	2.983	1.276	**<0.05**	0.852	0.814	NS
β-Cryptoxanthin	Model 1	−0.660	0.526	NS	0.163	1.044	NS	1.427	1.207	NS	0.672	0.336	<0.05	1.546	1.734	NS	0.403	1.102	NS
	Model 2	−0.236	0.494	NS	−0.773	0.962	NS	0.666	1.168	NS	0.131	0.229	NS	1.380	1.752	NS	0.578	1.112	NS
	Model 3	−0.301	0.501	NS	−0.983	0.973	NS	0.719	1.177	NS	0.141	0.231	NS	0.778	1.763	NS	0.578	1.116	NS
Lutein	Model 1	−0.530	0.675	NS	0.975	1.337	NS	2.091	1.545	NS	0.987	0.431	**<0.05**	−0.109	2.227	NS	0.673	1.412	NS
	Model 2	0.097	0.635	NS	−0.388	1.236	NS	0.987	1.501	NS	0.194	0.294	NS	−0.370	2.255	NS	0.941	1.427	NS
	Model 3	0.042	0.649	NS	−0.604	1.261	NS	1.524	1.522	NS	0.299	0.298	NS	−1.099	2.284	NS	1.437	1.443	NS
Zeaxanthin	Model 1	0.929	0.455	**<0.05**	0.528	0.907	NS	−0.595	1.051	NS	−0.413	0.294	NS	2.709	1.501	NS	−0.894	0.957	NS
	Model 2	0.617	0.427	NS	1.240	0.831	NS	−0.016	1.014	NS	−0.003	0.199	NS	2.878	1.512	NS	−1.034	0.962	NS
	Model 3	0.578	0.431	NS	1.139	0.837	NS	0.095	1.015	NS	0.019	0.199	NS	2.639	1.513	NS	−0.888	0.961	NS
Lycopene	Model 1	−0.578	0.819	NS	1.196	1.658	NS	1.594	1.866	NS	0.866	0.529	NS	0.026	2.613	NS	1.971	1.724	NS
	Model 2	−0.015	0.775	NS	−0.136	1.527	NS	0.434	1.788	NS	0.086	0.350	NS	−0.438	2.630	NS	2.243	1.737	NS
	Model 3	0.037	0.786	NS	−0.032	1.544	NS	0.711	1.782	NS	0.140	0.349	NS	−0.565	2.650	NS	2.634	1.737	NS
Total carotenoids	Model 1	−0.129	0.709	NS	0.798	1.404	NS	1.312	1.626	NS	0.616	0.456	NS	5.277	2.315	**<0.05**	0.628	1.483	NS
	Model 2	0.262	0.661	NS	−0.044	1.289	NS	0.617	1.565	NS	0.121	0.307	NS	5.163	2.328	**<0.05**	0.787	1.488	NS
	Model 3	0.143	0.676	NS	−0.422	1.313	NS	0.803	1.587	NS	0.157	0.311	NS	4.248	2.364	**<0.05**	0.997	1.504	NS

PACER, progressive aerobic cardiovascular endurance run; VO_2max_, maximal oxygen uptake. Total carotenoids: the sum of the blood levels of the six individual carotenoids. Model 1: adjusted for age. Model 2: adjusted for age and BMI. Model 3: adjusted for age, BMI, smoking, drinking, and physical activity.

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
