# Peer review of "Association of Carotenoids Concentration in Blood with Physical Performance in Korean Adolescents: The 2018 National Fitness Award Project"

_nutrients, 2020, doi:10.3390/nu12061821_

Round 1

Reviewer 1 Report

The paper “Association of Carotenoids Concentration in Blood with Physical Performance in Korean Adolescents: the 2018 National Fitness Award Project” describes the correlation between the carotenoids and the physical performance in 450 Korean adolescents aged 13 to 18 years.

The research is interesting but, in my opinion, few points should be improved before the publication.

The authors should explain acronyms the first time they use them and then continue to use them only as acronyms, for example BMI, PACER and estimated VO2max in abstract.

Data should be expressed with the same decimal unit, for example in table 1 there are data with one or two decimal number. Please, uniform it in all test.

Has a single test been done for each teenager or have the results been repeated in triplicate? Were carotenoid supplements administered to the teenagers who participated in the study? Did they follow a particular diet? Explain these points in the text, please.

The paper “Association of Carotenoids Concentration in Blood with Physical Performance in Korean Adolescents: the 2018 National Fitness Award Project” describes the correlation between the carotenoids and the physical performance in 450 Korean adolescents aged 13 to 18 years.

The research is interesting but, in my opinion, few points should be improved before the publication.

The authors should explain acronyms the first time they use them and then continue to use them only as acronyms, for example BMI, PACER and estimated VO2max in abstract.

Data should be expressed with the same decimal unit, for example in table 1 there are data with one or two decimal number. Please, uniform it in all test.

Has a single test been done for each teenager or have the results been repeated in triplicate? Were carotenoid supplements administered to the teenagers who participated in the study? Did they follow a particular diet? Explain these points in the text, please.

Author Response

The paper “Association of Carotenoids Concentration in Blood with Physical Performance in Korean Adolescents: the 2018 National Fitness Award Project” describes the correlation between the carotenoids and the physical performance in 450 Korean adolescents aged 13 to 18 years.

The research is interesting but, in my opinion, few points should be improved before the publication.

-> Authors: Thank you again for taking the time to review this paper. Please see our detailed responses below.

The authors should explain acronyms the first time they use them and then continue to use them only as acronyms, for example BMI, PACER and estimated VO2max in abstract.

-> Authors: Thank you for this suggestion; we have made the correction (Page 1, Lines 19-21).

Data should be expressed with the same decimal unit, for example in table 1 there are data with one or two decimal number. Please, uniform it in all test.

-> Authors: Normally, it is appropriate to display up to one decimal place. However, in the case of Table 2, the concentration values of carotenoids were lower than 1, so they were represented to two decimal places.

Has a single test been done for each teenager or have the results been repeated in triplicate? Were carotenoid supplements administered to the teenagers who participated in the study? Did they follow a particular diet? Explain these points in the text, please.

-> Authors: This study is a cross-sectional study, not a trial. Therefore, subjects did not take supplements and were not required to eat a special diet. At the time of physical performance measurement, the blood of the subjects was collected, the level of carotenoids in the blood was measured, and the relationship with physical performance was analyzed. We have added marks to this study design to the abstract and text sections. References to this research design have been added to the abstract and body sections (Pages 1&2&8, Lines 16&62&67&1).

Reviewer 2 Report

The Authors evaluated the association between the carotenoids concentration in blood with physical performance in Korean adolescents aged 13 to 18 years. This cross-sectional study has two serious limitations in my opinion:

1) the smoking, drinking individuals should be excluded before the data analysis (oxidative stress) and 2) the lack of nutritional assessment (the level of consumed carotenoids could affect the result).

As for the work addressed to the Nutrients it has relatively poor context to the food/diet.

Nevertheless the study is interesting and presents a new result for relation between the carotenoids (individuals and total) and physical performance.

Detailed comments:

In Abstract:

- PACER – abb. should be expanded

Introduction:

- It is not clarified as Authors indicated whether the physical activity increases antioxidant parameters (the data are full of discrepancies)

M & M:

The study sample is 450 subjects but current smokers and drinkers should not be included to the study group (possible bias of the whole analysis) - it is known that smoking intensifies oxidative stress (free radicals generation) and by this way it also affects the antioxidant status of body fluids. A number of cigarettes a day itself also strongly determines the level of oxidants and antioxidants.

In description of physical performance the past tense should be used.

2.6

…blood level ? of what?

Results

Tables 1, 2 and 4 (especially) are poorly readable when non-significant p values are numerical – the using of shortcut “NS” may improve the readability as well as “<0.05”, “<0.01”, “<0.001” instead of exact value of p.

The lack of nutritional data (carotenoids provided with food also could influence)

Discussion

Not only carotenoids are seen responsible for decreased oxidative stress.

Too many times authors admire their results as the student group in adolescence, a critical period, but they do not consider that the group consisted of the subjects at different developmental age 13-18 y – the group was representative but simultaneously this differentiation was a limitation of the study (different hormonal metabolism, especially in girls).

Conclusions:

1st and 2nd sentence describe the same thing.

Author Response

The Authors evaluated the association between the carotenoids concentration in blood with physical performance in Korean adolescents aged 13 to 18 years. This cross-sectional study has two serious limitations in my opinion:

1) the smoking, drinking individuals should be excluded before the data analysis (oxidative stress) and 2) the lack of nutritional assessment (the level of consumed carotenoids could affect the result).

As for the work addressed to the Nutrients it has relatively poor context to the food/diet.

Nevertheless the study is interesting and presents a new result for relation between the carotenoids (individuals and total) and physical performance.

-> Authors: We sincerely appreciate your insightful and constructive comments and suggestions. Please see our detailed responses below.

Detailed comments:

In Abstract:

- PACER – abb. should be expanded

-> Authors: Thank you for this suggestion. We added explanations when abbreviations (PACER) were first used (Page 1, Line 20).

Introduction\

- It is not clarified as Authors indicated whether the physical activity increases antioxidant parameters (the data are full of discrepancies)

-> Authors: Following your suggestion, we have revised the sentence to clarify this (Page 1, Line 44).

M & M:

The study sample is 450 subjects but current smokers and drinkers should not be included to the study group (possible bias of the whole analysis) - it is known that smoking intensifies oxidative stress (free radicals generation) and by this way it also affects the antioxidant status of body fluids. A number of cigarettes a day itself also strongly determines the level of oxidants and antioxidants.

-> Authors: Thank you for these valuable comments. We agree with you that drinking and smoking affect oxidative stress. However, even in cross-sectional studies involving other people explaining in connection with antioxidant mechanisms, most of them are analyzed by adding them to covariates rather than deleting smokers or drinkers altogether. In addition, the subjects of this study were middle and high school students, and those who smoked (one cigarette) or drank (one drink) in the past month were classified as drinkers and smokers, with 2.9% and 8.9%, respectively. We did not exclude these subjects from the analysis because less than 1% of respondents said they smoked or drank heavily. Instead, smoking and drinking variables were considered as covariates, and further analysis was conducted in model 3. The results were revised in the text, and the results of the model that adjusted both smoking and drinking were similar to those of the previous one (Table 4).

In description of physical performance the past tense should be used.

-> Authors: We appreciate your kind comments. In response to your suggestion, we attempted to revise the description section of physical performance to the past tense, but the sentence was not natural, so we kept the present tense. However, this part has been carefully revised once again as a whole by native speakers. Please understand (Pages 3&4, Lines 111-147).

2.6

…blood level ? of what?

-> Authors: We have modified this to the blood levels of carotenoids (Page 4, Lines 153).

Results

Tables 1, 2 and 4 (especially) are poorly readable when non-significant p values are numerical – the using of shortcut “NS” may improve the readability as well as “<0.05”, “<0.01”, “<0.001” instead of exact value of p.

-> Authors: Thank you for this suggestion. Based on your suggestion, we have modified the p values in the tables to improve the readability.

The lack of nutritional data (carotenoids provided with food also could influence)

-> Authors: Carotenoids are components that are not made by our body, and carotenoid levels in the blood are affected only by food. That is why we conducted this study on the assumption that blood carotenoid levels are positively related to the amount of carotenoids consumed through food. In other words, high blood carotenoid levels mean that you have consumed a large amount of carotenoids. Previous studies have already reported that there is a positive relationship between carotenoid intake and blood carotenoid levels. We further described the absence of nutritional assessments in the limitations section of the study (Page 9, Lines 71-73).

Discussion

Not only carotenoids are seen responsible for decreased oxidative stress.

-> Authors: The comment is most appreciated. We have modified this section so that it does not appear that only carotenoids are responsible for reducing oxidative stress (Page 8, Lines 27-28).

Too many times authors admire their results as the student group in adolescence, a critical period, but they do not consider that the group consisted of the subjects at different developmental age 13-18 y – the group was representative but simultaneously this differentiation was a limitation of the study (different hormonal metabolism, especially in girls).

-> Authors: Thank you for this comment. We added the content of the points you pointed out to the limitations of the study (Page 9, Lines 72-75).

Conclusions:

1st and 2nd sentence describe the same thing.

-> Authors: Thank you for this comment. We modified the first sentence to clearly state which carotenoids were associated with which type of physical performance (Page 9, Lines 83-85).

Reviewer 3 Report

  1. Why did you measure only carotenoid levels in blood? There might be other antioxidants in blood also contributed to physical performance. You should at least measure total antioxidant capacity in blood and evaluate its association with physical performance and discuss that with the association of carotenoids. How about effects of other nutrients – any interferences to the results?
  2. What were the controls in your study?
  3. Have there been any reports on the association of carotenoids and physical performance in other age groups in the literature? This needs to be compared with your results and discuss in the manuscript.
  4. Why were levels of lutein and zeaxanthin significantly different between boys and girls but there was no association of these carotenoids with physical performance? In contrast, there was no significant difference in levels of a-carotene and b-carotene but these carotenoids were associated with physical performance. What could be the reasons for this?
  5. You mentioned your work’s limitations. What are your suggestions to overcome these limitations and improve research outcome in future studies? Need to include this in your discussion.
  6. Introduction: provide more specific evidences on the relationship between plasma antioxidant capacity and physical performance.
  7. Conclusion: need to mention clearly which carotenoids was associated with which type of physical performance.
  8. Repetitive phrases: “which is a critical period, as the physical performance level in adolescence can be linked to post-adult health conditions or diseases” – appeared on page 2, 8, 9.

Author Response

1.Why did you measure only carotenoid levels in blood? There might be other antioxidants in blood also contributed to physical performance. You should at least measure total antioxidant capacity in blood and evaluate its association with physical performance and discuss that with the association of carotenoids. How about effects of other nutrients – any interferences to the results?

-> Authors: Thank you for your suggestion. Our lab focuses on phytochemicals, especially carotenoids, to study their relationship with health. Therefore, we first analyzed carotenoid levels in blood. We agree that the measurement of total antioxidant capacity in blood is critical for this research. However, due to the limited number of samples, we were not able to analyze total antioxidant capacity in blood. However, we have added a sentence regarding this aspect to the study limitations section (Page 9, Lines 68-70). We will consider the measurement of total antioxidant capacity in a future study.

2.What were the controls in your study?

-> Authors: This study is a cross-sectional study and does not have a control group. We analyzed the relationship between carotenoid levels in the blood and physical performance in one group at this time.

3.Have there been any reports on the association of carotenoids and physical performance in other age groups in the literature? This needs to be compared with your results and discuss in the manuscript.

-> Authors: Although unrelated to adolescents, the importance of maintaining physical performance in older adults is being emphasized as the elderly population increases worldwide, and the role of carotenoids [23] to prevent sarcopenia associated with physical performance in old age has been revealed. A study of American elderly individuals reported a relationship between low serum carotenoids and a decline in walking speed in older women [24], and a study in Italy revealed that skeletal muscle strength decreases over time (6 years) in older adults with low plasma carotenoid levels [25]. As such, previous studies have reported that carotenoids are directly or indirectly related to physical performance in the elderly. (Page 8, Lines 7-14).

4.Why were levels of lutein and zeaxanthin significantly different between boys and girls but there was no association of these carotenoids with physical performance? In contrast, there was no significant difference in levels of a-carotene and b-carotene but these carotenoids were associated with physical performance. What could be the reasons for this?

-> Authors: We do not think that carotenoids should be related to physical performance when analyzing boys and girls, respectively, as blood levels of carotenoids (lutein, zeaxanthin) differed in boys and girls. Table 1 simply compares blood carotenoid levels between boys and girls, and Table 4 presents the results of an analysis of the relationship between blood carotenoid levels and physical performance in boys and girls, respectively. The results of this study mean that the levels of a-carotene, b-carotene, and total carotenoids did not show a significant difference between boys and girls, but when analyzed by boys and girls, these carotenoid levels were positively related to physical performance in both boys and girls. This is the first study to hypothesize that carotenoid levels are related to physical performance, and further studies are needed to examine these associations in more depth.

5.You mentioned your work’s limitations. What are your suggestions to overcome these limitations and improve research outcome in future studies? Need to include this in your discussion.

-> Authors: Thank you for your suggestion. We included this part in the discussion section. (Page 9, Lines 74-76).

6.Introduction: provide more specific evidences on the relationship between plasma antioxidant capacity and physical performance.

-> Authors: As far as we know, there has been no direct study on the relationship between plasma antioxidant capacity and physical performance in healthy people (i.e., not patients). However, studies have shown that blood levels of vitamins with antioxidant capacity are positively related to physical performance. We added this to the introduction (Page 2, Lines 45-47).

7.Conclusion: need to mention clearly which carotenoids was associated with which type of physical performance.

-> Authors: As you suggested, we clearly mentioned which carotenoids were associated with which type of physical performance (Page 9, 83-85).

8.Repetitive phrases: “which is a critical period, as the physical performance level in adolescence can be linked to post-adult health conditions or diseases” – appeared on page 2, 8, 9.

-> Authors: We deleted the sentence on page 8.

Round 2

Reviewer 3 Report

The manuscript has been adequately revised.